# Protein Disulfide Isomerase and Extracellular Adherence Protein Cooperatively Potentiate Staphylococcal Invasion into Endothelial Cells

Marleen Leidecker,[a] Anne Bertling,[b] Muzaffar Hussain,[a] Markus Bischoff,[c] Johannes A. Eble,[d] Anke C. Fender,[b,e] Kerstin Jurk,[b,f] Christine Rumpf,[a] Mathias Herrmann,[a] Beate E. Kehrel,[b] Silke Niemann[a]

[a]Institute of Medical Microbiology, University Hospital of Münster, Münster, Germany
[b]Department of Anaesthesiology and Intensive Care, Experimental and Clinical Haemostasis, University Hospital of Münster, Münster, Germany
[c]Institute of Medical Microbiology and Hygiene, Saarland University, Homburg, Germany
[d]Institute of Physiological Chemistry and Pathobiochemistry, University of Münster, Münster, Germany
[e]Institute of Pharmacology, University Hospital Essen, Essen, Germany
[f]Center for Thrombosis and Hemostasis, University Medical Center of the Johannes Gutenberg-University Mainz, Mainz, Germany

Marleen Leidecker and Anne Bertling contributed equally to this work. Author order was determined by the chronological order of the work in the study; Marleen Leidecker took over the work from Anne Bertling.
Beate E. Kehrel and Silke Niemann also contributed equally to this work as senior authors.

**ABSTRACT** Invasion of host cells is an important feature of *Staphylococcus aureus*. The main internalization pathway involves binding of the bacteria to host cells, e.g., endothelial cells, via a fibronectin (Fn) bridge between *S. aureus* Fn binding proteins and $\alpha_5\beta_1$-integrin, followed by phagocytosis. The secreted extracellular adherence protein (Eap) has been shown to promote this cellular uptake pathway of not only *S. aureus*, but also of bacteria otherwise poorly taken up by host cells, such as *Staphylococcus carnosus*. The exact mechanisms are still unknown. Previously, we demonstrated that Eap induces platelet activation by stimulation of the protein disulfide isomerase (PDI), a catalyst of thiol-disulfide exchange reactions. Here, we show that Eap promotes PDI activity on the surface of endothelial cells, and that this contributes critically to Eap-driven staphylococcal invasion. PDI-stimulated $\beta_1$-integrin activation followed by increased Fn binding to host cells likely accounts for the Eap-enhanced uptake of *S. aureus* into non-professional phagocytes. Additionally, Eap supports the binding of *S. carnosus* to Fn-$\alpha_5\beta_1$ integrin, thereby allowing its uptake into endothelial cells. To our knowledge, this is the first demonstration that PDI is crucial for the uptake of bacteria into host cells. We describe a hitherto unknown function of Eap—the promotion of an enzymatic activity with subsequent enhancement of bacterial uptake—and thus broaden mechanistic insights into its importance as a driver of bacterial pathogenicity.

**IMPORTANCE** *Staphylococcus aureus* can invade and persist in non-professional phagocytes, thereby escaping host defense mechanisms and antibiotic treatment. The intracellular lifestyle of *S. aureus* contributes to the development of infection, e.g., in infective endocarditis or chronic osteomyelitis. The extracellular adherence protein secreted by *S. aureus* promotes its own internalization as well as that of bacteria that are otherwise poorly taken up by host cells, such as *Staphylococcus carnosus*. In our study, we demonstrate that staphylococcal uptake by endothelial cells requires catalytic disulfide exchange activity by the cell-surface protein disulfide isomerase, and that this critical enzymatic function is enhanced by Eap. The therapeutic application of PDI inhibitors has previously been investigated in the context of thrombosis and hypercoagulability. Our results add another intriguing possibility: therapeutically targeting PDI, i.e., as a candidate approach to modulate the initiation and/or course of *S. aureus* infectious diseases.

Address correspondence to Silke Niemann, silke.niemann@uni-muenster.de, or Beate E. Kehrel, kehrel@uni-muenster.de.

The authors declare no conflict of interest.

**KEYWORDS** *Staphylococcus aureus* host cell invasion, extracellular adherence protein, protein disulfide isomerase

*S*taphylococcus aureus is a major pathogen responsible for a wide range of infections, such as skin and soft tissue infections, endocarditis, pneumonia, and osteomyelitis (1). The bacterium can invade and persist in non-professional phagocytes, thereby escaping host defense mechanisms and antibiotic treatment (2–5). Uptake of *S. aureus* by host cells represents an important step, particularly in acute endocarditis, but probably also for the development of chronic infections such as chronic osteomyelitis. (6–9).

Uptake of *S. aureus* into non-professional phagocytes is mediated primarily through fibronectin-binding proteins (FnBPs) on the pathogen cell surface, by fluid-phase fibronectin (Fn) as bridging molecules, and by $\alpha_5\beta_1$ integrin on the cell surface of host cells (10). Binding of *S. aureus* to the host cell leads to the clustering of $\alpha_5\beta_1$ integrin, which in turn triggers intracellular signaling cascades that subsequently lead to a restructuring of the actin cytoskeleton and mobilization of endocytosis machinery (10–14). *S. aureus* internalization is further potentiated via the functional interaction of certain bacterial surface structures, such as autolysin (Atl) and lipoproteins, with the host cognate heat shock cognate protein 70 and heat shock protein 90, respectively (15–18).

As far as is known, the cell surface-associated extracellular adherence protein (Eap) is a unique feature of the species *S. aureus* and *Staphylococcus argenteus* (19, 20) and contributes to the high virulence of these pathogens. Eap is a highly cationic protein that belongs to the so-called SERAM (secretable expanded repertoire adhesive molecules) group (21). These secreted bacterial proteins bind to host adhesive factors and/or mediate bacterial adhesion to host molecules, cells, or tissues, and interact with a wide range of host ligands, thereby impairing host defense mechanisms. Eap lacks the typical LPXTG motif that mediates protein anchoring to the bacterial cell wall. However, a significant portion of secreted Eap can reassociate with the bacterial surface (21). Eap has various immunomodulatory properties, such as inhibiting the classical and lectin pathways of complement, the binding of neutrophils to endothelial cells, the formation of extracellular traps for neutrophils, and the degradation of phenol-soluble modules of *S. aureus* by neutrophil serine proteases (22–25). Eap also exhibits an extensive binding repertoire to matrix and plasma proteins (26, 27). This protein was described to promote bacterial adherence (27, 28) and invasion (29, 30) by way of its dual affinity for both eukaryotic elements and the bacterium itself (27, 29). However, host cell receptors assisting Eap-mediated internalization in the endothelium have not been previously defined. *S. aureus* Eap also promotes the invasion capacity of other bacteria, namely, *Staphylococcus carnosus*, *Staphylococcus epidermidis*, and *Staphylococcus lugdunensis*, which are themselves not capable of Eap production and are otherwise only poorly taken up by non-professional phagocytes (29, 30).

Previously, we identified certain members of the thiol isomerase family, specifically the protein disulfide isomerase (PDI, also known as P4HB), as important functional partners for Eap-induced platelet activation and aggregation. Notably, we demonstrated that Eap can bind directly to PDI (31). Although the PDI is typically localized in the lumen of the endoplasmic reticulum (32–34), externalized PDI has also been described on the surface of different cell types, including endothelial cells (34–37). PDI acts as both a catalyst of thiol-disulfide exchange reactions and as a chaperone protein (33, 34, 38). It has been shown that the PDI modulates the function of integrins such as $\alpha_{IIb}\beta_3$ (39, 40), $\alpha_\nu b_3$ (41), $\beta_1$ (42, 43), and $\alpha_5\beta_1$ (44) and, moreover, is involved in the integrin-mediated adhesion of cells to Fn and collagen (44). This constellation of activities implicates membrane-associated PDI in the pathogenesis of diverse infectious diseases caused by HIV (45, 46), *Chlamydia* (47, 48), and dengue virus (43), as well as in the contexts of cancer (36) and cardiovascular diseases (49). A supportive role of PDI in Eap-driven *S. aureus* internalization has not previously been reported. In this study, we demonstrate that *S. aureus* uptake by endothelial cells requires the catalytic disulfide exchange activity of PDI on the cell surface and that this critical enzymatic function is enhanced by Eap.

## RESULTS

**Eap-dependent adhesion and internalization of _S. aureus_ and _S. carnosus_.** Because earlier studies have already described a strain-dependent augmentation of extracellular Eap on the adhesion and internalization of bacteria into host cells (29), we tested the adhesion and internalization rates of three staphylococcal strains with differing endogenous Eap expression rates: _S. carnosus_ TM300 (does not produce Eap), _S. aureus_ 8325-4 (a low-level Eap-producing strain), and _S. aureus_ Newman (a high-level Eap-producing strain) (50, 51). The adhesion of _S. aureus_ 8325-4 to unstimulated HMEC-1 endothelial cells (EC) was approximately 3-fold that of _S. carnosus_ TM300 (Fig. 1A and B; absolute numbers are shown in Fig. S1A and B), and following pretreatment of HMEC-1 with exogenous Eap, only the adherence of _S. carnosus_ TM300 was notably increased. Internalization of _S. aureus_ 8325-4 into unstimulated HMEC-1 was approximately 85-fold that of _S. carnosus_ TM300 (Fig. S1C and D); in Eap-pretreated HMEC-1, uptake of _S. aureus_ 8325-4 was augmented by 50%, while massively increased uptake (up to around 1,500%) was noted for _S. carnosus_ TM300 (Fig. 1C and D, Fig. S1E). The same patterns for Eap-regulated internalization were observed in the EA.hy926 cell line (Fig. S1F to J). To validate the critical role of Eap in _S. aureus_ uptake, we compared the Eap-producing _S. aureus_ strain Newman (50) and the _eap_-deficient mutant strain _S. aureus_ Newman mAH12 (28). Internalization of _S. aureus_ mAH12 by HMEC-1 was marginal compared to that of the wild-type strain but could be rescued with addition of exogenous Eap (Fig. 1E).

**Eap promotes protein disulfide isomerase activity on the surface of endothelial cells.** Exposure of endothelial cells (HMEC-1 and EA.hy926) to Eap led to a significant and concentration-dependent increase in the abundance of cell-surface free ecto-sulfhydryls, a measure of PDI activity, as assessed by binding of the thiol-reactive dye Alexa Fluor 488 $C_5$ Maleimide to the EC surface (Fig. 2A and B, Fig. S2A). The Eap-driven PDI activity in HMEC-1 was diminished by treatment with either P4HB (PDI) small interfering RNA (siRNA) or the inhibitory anti-PDI monoclonal antibody RL90 (Fig. 2C). Similar observations were made in EA.hy926 cells (Fig. S2B).

Conditioned supernatant (sterile-filtered) from an overnight culture of _S. aureus_ Newman increased levels of free sulfhydryls on the surface of HMEC-1; this effect was significantly less pronounced with conditioned supernatant from _S. aureus_ mAH12, and completely absent with supernatant from _S. aureus_ 8325-4 and _S. carnosus_ TM300 (Fig. 2D). The thiol isomerase activity of soluble PDI, as assessed by oxidative refolding of scrambled RNase A, was markedly augmented in the presence of Eap (Fig. S2C), as was reduction of di-eosin-oxidized-glutathione (di-E-GSSG) to E-GSH, a measure of PDI reductase activity, both in a cell-free system (Fig. 2E, Fig. S2D) and on live cells (Fig. 2F). The PDI inhibitor bacitracin, a membrane-impermeable antibiotic targeting the substrate-binding domain of PDI (52), and the thiol blocker 5,5′-dithiobis-(2-nitrobenzoic acid) (DTNB) attenuated Eap-stimulated PDI reductase activity on the surface of HMEC-1 cells, as did P4HB siRNA and the inhibitory anti-PDI monoclonal antibody RL90 (Fig. 2G). Eap alone, in the absence of PDI, did not modify E-GSH formation in a cell free system, nor did it stimulate the thiol isomerase activity of Fn (both Fig. S2D).

**PDI contributes to Eap-driven staphylococci invasion in eukaryotic cells.** To investigate the contribution of thiol isomerases to the Eap-driven internalization process of bacteria into host cells, we examined the effect of PDI-inhibiting compounds on staphylococcal uptake.

The conventional PDI inhibitor bacitracin markedly reduced Eap-driven bacterial uptake by EA.hy926 (Fig. 3A and B) and HMEC-1 cells (Fig. S3A and B). Both bacitracin and any unbound Eap were washed away before the ECs were incubated with the bacteria. Interestingly, we observed an inhibitory effect of bacitracin on basal _S. aureus_ uptake in the absence of exogenous Eap (Fig. 3A, Fig. S3A).

Similar inhibition of basal and Eap-stimulated EC invasion was observed with the membrane-impermeable thiol blocker DTNB. The uptake of bacteria was suppressed in a particularly strong manner (Fig. 3C and D, Fig. S3C and D).

The highly cell-permeant and selective PDI inhibitor rutin (53) did not influence basal internalization of _S. aureus_ in the absence of Eap, but markedly suppressed Eap-

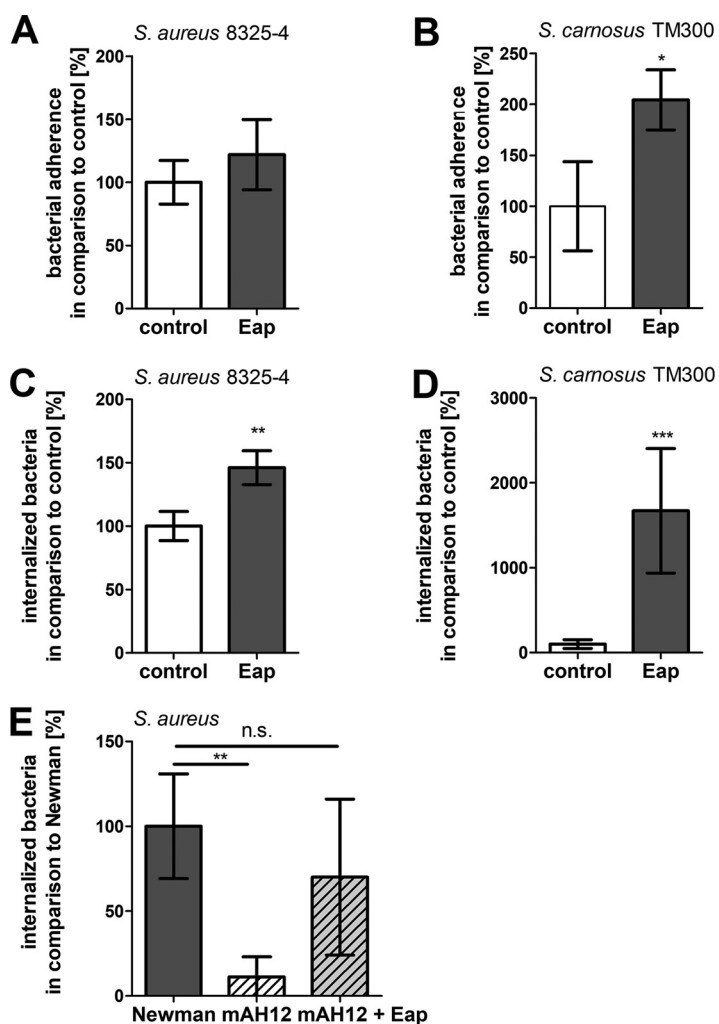

**FIG 1** Extracellular adherence protein (Eap) enhances staphylococcal adherence to and internalization in endothelial cells. (A) Adhesion of *Staphylococcus aureus* strain 8325-4 and (B) *Staphylococcus carnosus* TM300 to HMEC-1 cells in the absence and presence of 20 $\mu$g/mL recombinant Eap. Cells were incubated with a multiplicity of infection (MOI) of 50. One hour postinfection, unbound bacteria were washed away with phosphate-buffered saline (PBS). Host cells were lysed with ice-cold distilled water to release extra- and intracellular bacteria, and the number of bacteria was assessed by plate counting. Bacterial numbers in control cells were set to 100%. Data represent the means ± standard deviation (SD) of four (*S. aureus*) or three (*S. carnosus*) independent experiments. *, $P < 0.05$, unpaired *t* test. (C) Internalization of *S. aureus* strain 8325-4, (D) *S. carnosus* TM300, or (E) *S. aureus* Newman and *S. aureus* mAH12 in HMEC-1 cells in the absence and presence of 20 $\mu$g/mL recombinant Eap. Cells were incubated with an MOI of 50. One hour postinfection, extracellular staphylococci were removed by lysostaphin treatment. Host cells were lysed with ice-cold distilled water in order to release intracellular bacteria, and the number of bacteria was assessed by plate counting. Bacterial numbers in control cells were set to 100%. Data represent the means ± SD from five independent experiments. ***, $P \leq 0.001$; **, $P \leq 0.01$, unpaired *t* test (A to D); data represent the means ± SD from four independent experiments. **, $P \leq 0.01$; n.s., not significant, one-way analysis of variance (ANOVA) followed by Bonferroni post-test (E).

stimulated uptake of *S. aureus* and *S. carnosus* (Fig. S3E and F). The solvent vehicle for DTNB and rutin (up to 1% [vol/vol] dimethyl sulfoxide [DMSO]) had only negligible direct effects on endothelial uptake of *S. aureus* (Fig. 3C, Fig. S3C and E) or *S. carnosus* TM300 (Fig. 3D and E, Fig. S3D and F).

The participation of endothelial surface PDI in staphylococcal internalization was verified with the inhibitory anti-PDI monoclonal antibodies RL77 and RL90. Both anti-PDI antibodies significantly attenuated the Eap-driven bacterial invasion of *S. carnosus*, whereas the isotype control IgG had no effect (Fig. 3F and G). Considering the possible interaction of *S. aureus* protein A with the inhibitory antibodies, we refrained from

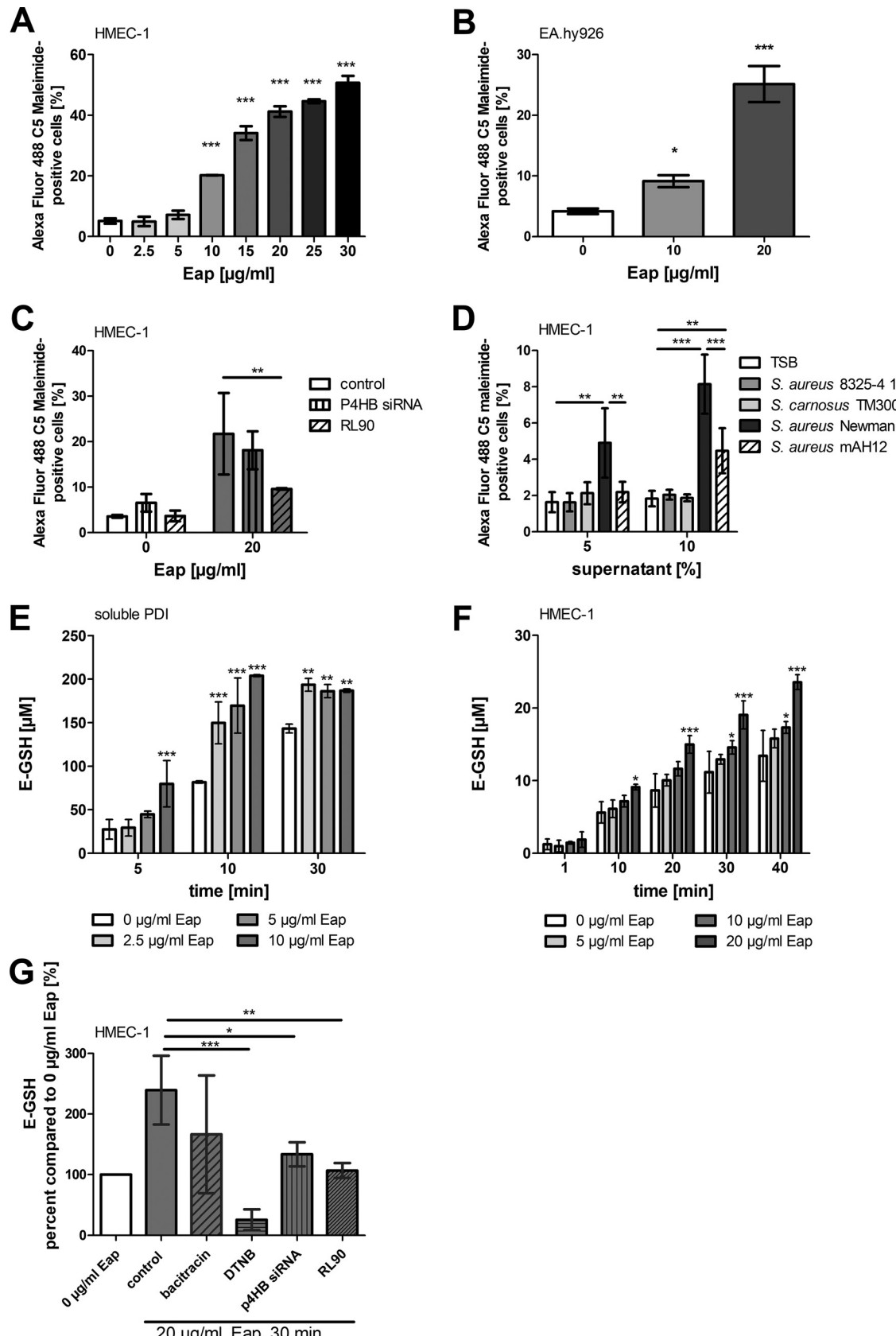

**FIG 2** Eap promotes protein disulfide activity on the surface of eukaryotic cells. (A) Eap-stimulated abundance of free ecto-sulfhydryls on the surface of HMEC-1 and (B) EA.hy926 cells was detected with the thiol-reactive dye Alexa Flour 488 $C_5$ Maleimide

antibody-based experiments with *S. aureus* to preclude nonspecific internalization effects. Transfection of EA.hy926 with P4HB siRNA also resulted in a significant decrease in Eap-stimulated uptake of *S. aureus* and *S. carnosus* into EC (Fig. 3H and I). Figure 3J shows the successful siP4HB knock down in EA.hy926 cells.

**Eap-driven activation of integrin $\beta_1$ and implications for Fn binding.** A relationship between PDI and integrin activation has been described (44, 54–56). The $\alpha_5\beta_1$ integrin is particularly important for the uptake of *S. aureus* into host cells (57, 58). Therefore, we used a flow-cytometry set-up to test whether Eap has an effect on $\beta_1$-integrin activation in HMEC-1 and EA.hy926 cells. Eap at concentrations up to 20 $\mu$g/mL had only modest effects in this regard, but administration of 50 $\mu$g/mL Eap resulted in a markedly increased binding of the antibody directed against activated $\beta_1$-integrin (Fig. 4A and B, Fig. S4A). This activation was inhibited by PDI knockdown by P4HB siRNA, the anti-PDI antibody RL77, and the PDI inhibitor bacitracin (Fig. 4A and B, Fig. S4A).

To clarify whether the Eap-enhanced uptake of *S. carnosus* was integrin-mediated, the uptake of these bacteria was examined in the presence of the $\alpha_5\beta_1$-integrin antagonist ATN161 (59). This inhibitor significantly diminished Eap-enhanced uptake (Fig. S4B).

During the internalization process, *S. aureus* binds to the $\alpha_5\beta_1$ integrin via an Fn bridge (13, 58, 60). Consequently, the effect of Eap on Fn binding to HMEC-1 and EA.hy926 cells was investigated. Exposure of ECs to Eap promoted the binding of FITC (fluorescein-5-isothiocyanate)-coupled Fn in a concentration-dependent manner (Fig. 4C and D). The contribution of PDI in this process was confirmed by the significant inhibitory effect of bacitracin. Also, a P4HB siRNA knockdown resulted in decreased binding of Fn to host cells (Fig. 4C and D; Fig. S4C illustrates the successful siP4HB knockdown in HMEC-1).

Finally, we tested the interaction of exogenously added Eap and Fn on the subsequent uptake of *S. aureus* and *S. carnosus*, respectively. As shown previously, Eap enhanced the internalization of *S. aureus* and *S. carnosus*, while Fn alone only significantly increased the uptake of *S. aureus* in ECs. However, Eap and Fn in combination led to the highest uptake of both *S. aureus* and *S. carnosus* in both cell lines (Fig. 4E to H). The PDI inhibitor bacitracin significantly inhibited bacterial internalization prompted by either Eap alone or Eap in combination with Fn. Increased uptake of *S. aureus* by Fn was also prevented by bacitracin, while this was not the case for Fn-supplemented *S. carnosus* cells, which internalized into cells of this EC lineage with comparable efficacies in the presence and absence of bacitracin (Fig. 4E and F). A decrease in Eap- or combined Eap- and Fn-mediated internalization of *S. aureus* 8325-4 and *S. carnosus* TM300 into HMEC-1 cells was also achieved by siP4HB (PDI) knockdown. However, the effect was smaller than that caused by bacitracin (Fig. 4G and H).

## DISCUSSION

In this study, we demonstrate that the thiol isomerase PDI is involved in *S. aureus* uptake in non-professional phagocytes, presumably by influencing the activation state of $\beta_1$ integrin. The *S. aureus* SERAM Eap has a stimulatory effect on the PDI and thereby leads to increased binding of Fn to $\beta_1$ integrin and to an augmented uptake of *S. aur-*

**FIG 2** Legend (Continued)

and measured by flow cytometry. Recombinant Eap data represent the means $\pm$ SD of three independent experiments. ***, $P \leq 0.001$; *, $P < 0.05$; one-way ANOVA followed by Dunnett's multiple-comparison test. (C) Inhibition of Eap-promoted abundance of ecto-sulfhydryls on HMEC-1 by P4HB siRNA (small interfering RNA) knockdown (20 pM siRNA) or anti-PDI antibody RL90 (10 $\mu$g/mL) as detected with Alexa Flour 488 $C_5$ Maleimide. Native Eap data represent the means $\pm$ SD of three independent experiments. **, $P \leq 0.01$; ***, $P \leq 0.001$; two-way ANOVA followed by Bonferroni post-test. (D) Bacterial supernatant stimulated (protein disulfide isomerase) PDI activity on the surface of HMEC-1 cells as detected with Alexa Flour 488 $C_5$ Maleimide. Data represent the means $\pm$ SD of three independent experiments. **, $P \leq 0.01$; two-way ANOVA followed by Bonferroni post-test. (E) Influence of Eap on kinetics of di-E-GSSG (di-eosin-oxidized-glutathione) reduction to E-GSH catalyzed by soluble PDI (200 nM) or (F) *in situ*-activity in HMEC-1 determined by measurement of fluorescence. Recombinant Eap data represent the means $\pm$ SD of three independent experiments. ***, $P \leq 0.001$; **, $P \leq 0.01$; *, $P < 0.05$; two-way ANOVA followed by Bonferroni post-test, compared to 0 $\mu$g/mL Eap. (G) Impact of bacitracin (10 mM), 5,5'-dithiobis-(2-nitrobenzoic acid) (DTNB, 10 mM), anti-PDI antibody RL90 (10 $\mu$g/mL), or P4HB siRNA knockdown (20 pM siRNA) on Eap (20 $\mu$g/mL natives Eap) promoted di-E-GSSG reduction to E-GSH in HMEC-1 cells, determined by measurement of fluorescence. Data represent the means $\pm$ SD of four independent experiments. ***, $P \leq 0.001$; **, $P \leq 0.01$; *, $P < 0.05$; one-way ANOVA followed by Dunnett's multiple-comparison test (compared to 20-$\mu$g/mL control).

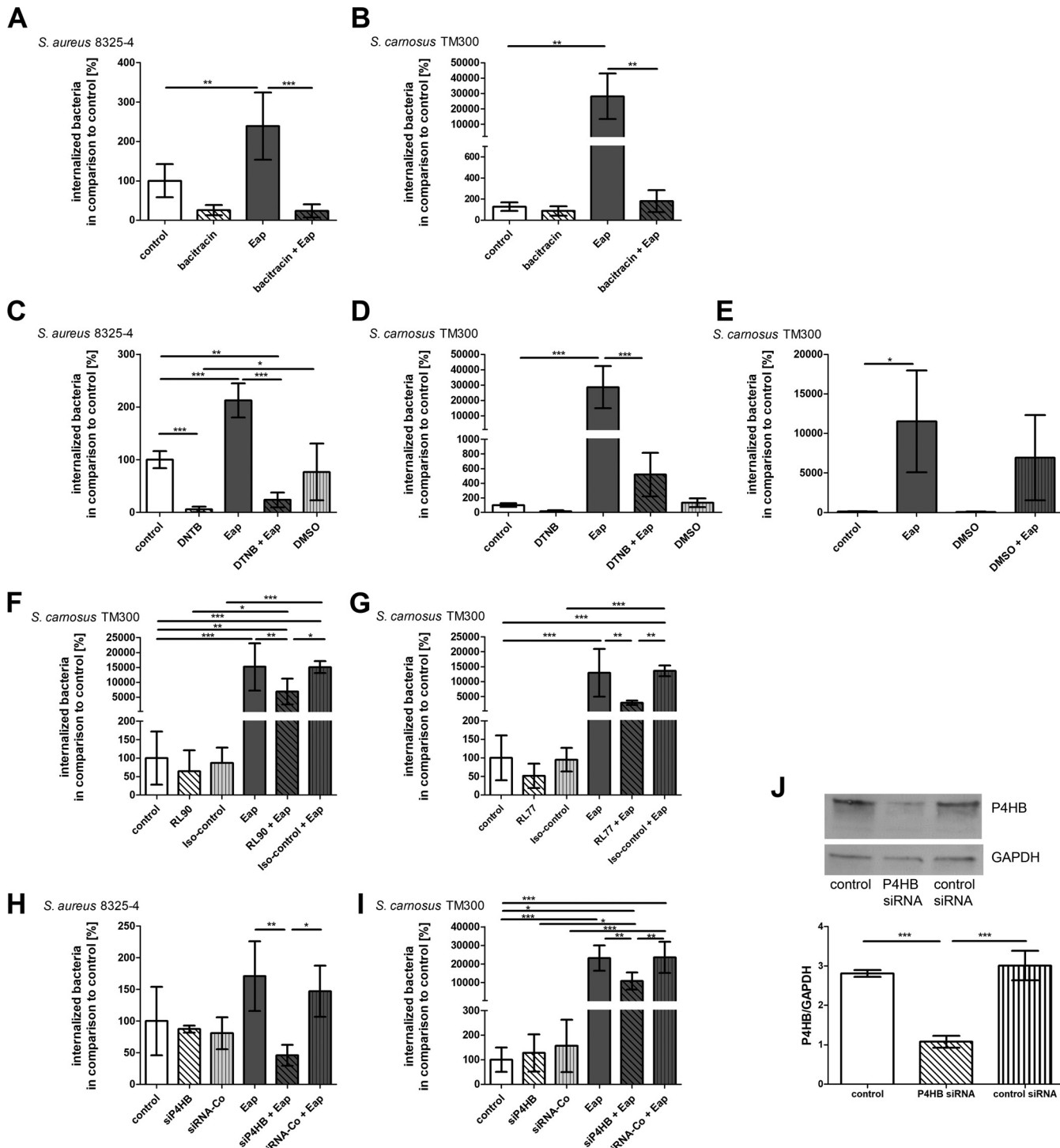

**FIG 3** Inhibition of protein disulfide isomerase reverses the enhancing effect of Eap on staphylococcal internalization. Internalization of *S. aureus* strain 8325-4 and *S. carnosus* TM300 as indicated in EA.hy926 cells in absence and presence of 20 $\mu$g/mL recombinant Eap. Cells were pre-incubated with (A and B) bacitracin (10 mM), (C and D) dithiobis-nitrobenzoic acid (DTNB, 10 mM), (E) DMSO (1% vol/vol), (F) anti-PDI antibody RL90 (10 $\mu$g/mL), or (G) anti-PDI antibody RL77 (1:100) 30 min before stimulation with Eap or (H and I) PDI expression silencing by P4HB siRNA knock down. In the case of bacitracin, host cells were washed before the addition of bacteria to remove bacitracin. Cells were incubated with an MOI of 50. One hour postinfection, extracellular staphylococci were removed by lysostaphin treatment. Intracellular bacteria numbers were assessed by plate counting. Bacterial numbers in control cells were set to 100%. Recombinant Eap data represent the means ± SD of at least three independent experiments. ***, $P \leq 0.001$; **, $P \leq 0.01$; *, $P < 0.05$; one-way ANOVA followed by Bonferroni post-test. (J) siRNA P4HB knockdown (10 pM siRNA) was confirmed by Western blotting. Data represent means ± SD from Western blot quantification of three independent experiments. ***, $P \leq 0.001$; one-way ANOVA followed by Bonferroni post-test.

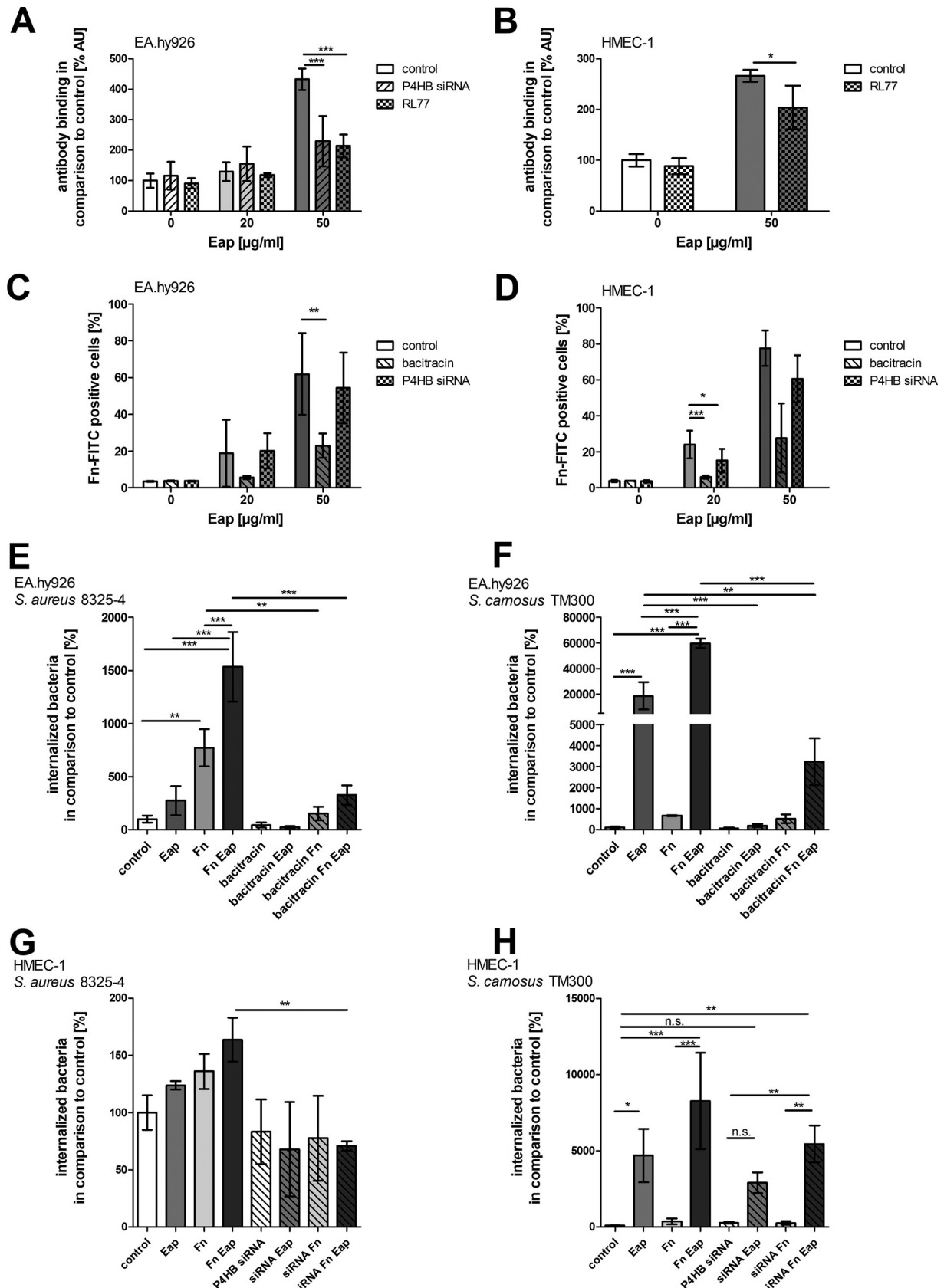

**FIG 4** Eap enhances activation of $\beta_1$ integrin and binding of Fn to cells in a thiol isomerase-dependent manner, affecting staphylococcal uptake. (A) Binding of FITC (fluorescein-5-isothiocyanate)-labeled antibody against activated anti-integrin $\beta_1$ to detached EA.h926 or (B) HMEC-1

*eus* into host cells. Notably, Eap also strongly enhances the uptake of cells of the weakly invasive species *S. carnosus* into ECs, and this effect is even further enhanced by the addition of Fn, presumably because it promotes the binding of *S. carnosus* TM300 to Fn-$\alpha_5\beta_1$ integrin complexes. To our knowledge, this is the first demonstration that PDI is critical for uptake of *S. aureus* into host cells. We describe a hitherto unreported function of Eap—promotion of a cell-surface enzymatic activity with subsequent enhancement of bacterial uptake—thereby advancing mechanistic insight into its importance as a driver of bacterial pathogenicity.

Internalization of *S. aureus* by non-professional phagocytes such as ECs is one of the ways this pathogen protects a part of its population from host defense mechanisms and antibiotic therapy, and in addition, contributes to the development of infection. The main internalization pathway utilized by *S. aureus* is the binding of its cell wall-anchored FnBPs to the $\alpha_5\beta_1$ integrin receptors on host cells via Fn as a bridging molecule, yet other staphylococcal factors also play a role in colonization processes (12). Exogenously added Eap, for instance, promotes bacterial adherence and invasion of/into ECs and fibroblasts, independently of endogenous Eap production (28, 30). Intriguingly, exogenous Eap also increases uptake of bacteria into non-professional phagocytes that cannot produce Eap themselves (29, 30), such as *S. carnosus*, implying support for other pathogens in multispecies infection scenarios.

In the present study, we used two different endothelial cell lines: the EA.hy926 cell line, conventionally used to investigate large-vessel endothelium, and the HMEC-1 cell-line as a model of microvascular endothelium (61). For most experiments, we used 20 µg/mL Eap. The amount of Eap isolated from the supernatant from a 24-h culture of *S. aureus* Newman was found to be 6.47 µg/mL (27). Thus, we used a slightly higher concentration for our experiments than that produced by *S. aureus* Newman, yet one of the same order of magnitude. We confirmed enhanced adherence and internalization of *S. aureus* strain 8325-4 to/into both EC lines upon external Eap addition, as well as strong augmentation of *S. carnosus* TM300 uptake into this host cell type. Notably, *S. aureus* 8325-4 is known to produce only small amounts of Eap by itself (50), which allowed us to particularly examine the influence of exogenously added Eap on *S. aureus* internalization in a "wild-type" FnBP background. The relevance of Eap in invasion was also demonstrated in an experiment performed with strain Newman, which expresses high levels of Eap, and its Eap-deficient mutant mAH12. The mutant was taken up to a significantly lesser extent than the wild-type strain in our experiments, with partial rescue noted with addition of exogenous Eap. Our data confirm those of Haggar et al. (30). However, in their study, uptake of the mAH12 strain after Eap addition was higher than that of the wild-type *S. aureus* Newman strain. This difference could be due to the different cell types used (HMEC-1 here, fibroblasts by Haggar et al.). In addition, the previous study used 80 µg/mL rather than 20 µg/mL Eap and the invasion time was 2 h rather than 1 h. *S. aureus* Newman and mAH12 were not used for further invasion assays because *S. aureus* Newman has truncated FnBPs that are not anchored in the cell wall. This leads to a loss of FnBP-dependent functions, such as strong adhesion to immobilized fibronectin, binding of fibrinogen, and invasion of host cells (62).

**FIG 4** Legend (Continued)
cells was determined by flow cytometry after incubation of cells with native Eap and/or anti-PDI antibody RL77 (1:100; added to the cells 15 min before addition of Eap) or P4HB siRNA knockdown; binding of antibody to control (no Eap, no inhibition) was set at 100%. Data represent the means ± SD of three independent experiments. ***, $P \leq 0.001$; *, $P < 0.05$; two-way ANOVA followed by Bonferroni post-test. (C) Binding of FITC-coupled Fn (Fn-FITC, 50 µg/mL) to EA.hy926 or (D) HMEC-1 cells in the absence or presence of native Eap, determined by flow cytometry. Cells were incubated with bacitracin (10 mM) 30 min before Eap incubation or transfected with P4HB siRNA (10 pM siRNA) 2 days before the experiment. Data represent the means ± SD of 3 independent experiments. ***, $P \leq 0.001$; **, $P \leq 0.01$; *, $P < 0.05$; two-way ANOVA followed by Bonferroni post-test. (E and G) Internalization of *S. aureus* 8325-4 or (F and H) *S. carnosus* TM300 in (E and F) EA.hy926 or (G and H) HMEC-1 cells after incubation with 20 µg/mL recombinant Eap, 50 µg/mL Fn, or both proteins together. (E and F) Bacitracin was added 30 min before the addition of Eap or Fn. Bacitracin, as well as unbound Eap or Fn, was washed away before cells were incubated with an MOI of 50. (G and H) Cells were transfected with P4HB siRNA (20 pM) 2 days before the experiment. One hour postinfection, extracellular staphylococci were removed by lysostaphin treatment. Intracellular bacteria numbers were assessed by plate counting. Numbers of bacteria in control cells were set to 100%. Data represent the means ± SD from three independent experiments. ***, $P \leq 0.001$; **, $P \leq 0.01$; *, $P < 0.05$; n.s., not significant; one-way ANOVA followed by Bonferroni post-test.

The uptake of *S. carnosus* under Eap impact was decreased by ATN161, an antagonist of integrin $\alpha_5\beta_1$ (and $\alpha_\nu\beta_3$), so we can assume that at least part of the Eap-mediated *S. carnosus* uptake is integrin-dependent.

We previously identified thiol isomerases as important interaction partners in the context of Eap-stimulated platelet activation and demonstrated that Eap can bind directly to PDI (31). These results, as well as the knowledge that PDI is found on the surface of ECs (34–37) and is involved in integrin activation (55), led us to investigate the role of PDI in the Eap-driven uptake of bacteria into this host cell type. We demonstrate the stimulation of PDI activity by Eap by different assays. Eap enhanced the thiol isomerase and reductase activities of PDI directly in a cell-free system, as well as on the surface of live EC, where it also increased the amount of free sulfhydryls. Eap itself does not contain cysteine residues (51), supporting the interaction between Eap and PDI on the eukaryotic cell surface. In our bacterium-cell interaction experiments, we mainly used the strains *S. aureus* 8325-4 and *S. carnosus* TM300. The low or absent Eap expression of these strains was also reflected by the fact that bacterial supernatants from these strains did not elicit thiol isomerase activity in HMEC-1 cells, in contrast to supernatant from *S. aureus* Newman, which is known to express large amounts of Eap. Incubating the cells with 10% supernatant of the Eap-negative mutant mAH12 significantly increased the binding of maleimide to cells, but the increase was significantly less than that by *S. aureus* Newman and could also be the result of increased cell death at this amount of supernatant. Our proposed link between Eap and PDI stimulation was also supported by the observation that the nonspecific thiol isomerase inhibitor DTNB, as well as PDI siRNA and the use of a PDI-blocking antibody, significantly inhibited the reductase activity in cells upon Eap incubation. Consistent with these results, the PDI-blocking antibody also inhibited the Eap-dependent increase in the amount of free sulfhydryls on the cell surface.

The membrane-impermeable, non-selective thiol isomerase inhibitor bacitracin markedly reduced the uptake of *S. aureus* and *S. carnosus* by ECs, and this effect was observable in the presence and absence of Eap. Since bacitracin also functions as an antibiotic, free bacitracin was removed before *S. aureus* cells were added. Of note, in the experimental setup with bacitracin, Eap was removed along with the bacitracin, confirming earlier findings that unbound Eap is not required for Eap-driven bacterial invasion of non-professional phagocytic cells (29). Internalization of *S. aureus* by ECs was also considerably reduced after treatment of the host cells with DTNB, a membrane-impermeable thiol-blocking compound. Again, the inhibitory effect of DTNB was seen in both the presence and absence of Eap, implying a general role of free exofacial thiol groups in *S. aureus* uptake. Additionally, inhibition of PDI by the PDI-selective compound rutin, by inhibitory anti-PDI antibodies, and by knockdown of PDI all reversed the enhancing effect of Eap on bacterial uptake by ECs. In summary, the nonspecific thiol isomerase inhibitors strongly inhibited uptake in the absence and presence of Eap, suggesting that thiol isomerases on the surface of endothelial cells are essential for the internalization of staphylococci by ECs. The fact that the degree of inhibition by PDI-selective inhibitors was lower than that by non-selective inhibitors, particularly the fact that the enhancement of uptake by Eap was reversed, suggests that PDI plays an important role in the Eap-driven internalization process of bacteria by ECs. Noteworthy, earlier work has demonstrated that Fn, which has a key role in *S. aureus* uptake by non-professional phagocytic cells via bridging staphylococcal FnBPs and host surface $\alpha_5\beta_1$ integrins, also inherits an intrinsic PDI activity (63). However, because we did not detect increased PDI reductase activity of Fn in response to Eap, we assume that this activity is not relevant to the Eap-driven internalization process of bacteria by ECs. However, we cannot exclude that the high level of bacterial uptake by ECs in the presence of Eap and Fn may be at least partially due to the intrinsic PDI activity of Fn.

It is well known that PDI has an important role in integrin activation (39, 44, 55, 56). We showed here that the addition of Eap to HMEC-1 cells also caused an increased activation of $\beta_1$-integrin, although this effect was more evident at high concentrations (i.e., 50 $\mu$g/mL) of this bacterial adhesion protein. However, cell culture conditions at

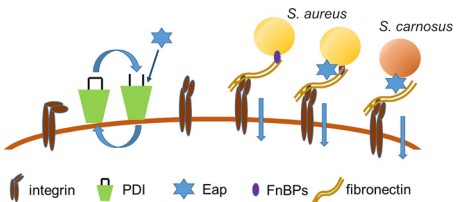

**FIG 5** Hypothetical model of how Eap mediates the internalization of *S. aureus* and other bacteria into host cells. Integrin $\alpha_5\beta_1$ on the host cell is not in an active state. Eap stimulates PDI, which leads to increased activation of $\alpha_5\beta_1$ integrins. This enhances the binding of Fn. *S. aureus* binds to Fn via Fn-binding proteins (FnBPs). Eap binds to both *S. aureus* and Fn and therefore, in concert with FnBPs, may facilitate the binding of *S. aureus* to $\alpha_5\beta_1$-integrin via the Fn bridge. Enhanced binding of Fn leads to increased integrin clustering, which results in elevated uptake of *S. aureus* into the host cell. In the case of *S. carnosus* and also other bacteria that are normally only poorly taken up by host cells, the increased integrin clustering caused by Eap presumably induces a more vivid internalization activity of ECs, accompanied by Eap bridges between the bacterium and Fn bound to $\alpha_5\beta_1$-integrin that enables the bacterial uptake.

atmospheric $O_2$ concentration represent a more oxidative environment than that which occurs in tissue *in situ*. Also, the sensitivity of either the antibody or the method might be responsible for the higher Eap concentration requirement. Importantly, this activation could be inhibited by bacitracin, by the PDI-inhibiting antibody RL77, and by P4HB siRNA knockdown, suggesting that PDI is involved in this activation. This further supports our hypothesis that Eap stimulates PDI, which in turn has an activating effect on integrins. The activation of the $\beta_1$-integrin may cause enhanced binding of, e.g., Fn to the integrin. In line with this assumption, we showed that the binding of FITC-labeled Fn to ECs was significantly promoted when they were stimulated with Eap. The contribution of PDI to this process was evidenced by the fact that the increased binding of Fn due to Eap could be significantly reduced by bacitracin or by P4HB siRNA knockdown. Eap also binds to cells via extracellular matrix proteins other than Fn and could bind Fn itself (64). Therefore, it cannot be excluded that part of the Fn binding may be due to a direct interaction with Eap fractions bound to the ECs. In the context of bacterial internalization, the addition of Fn resulted in enhanced uptake rates, and this effect was further intensified in the presence of Eap. Regarding *S. carnosus*, which does not produce a factor with an affinity for Fn, there was a weak but non-significant effect of Fn on host cell internalization. In contrast, when cells were pre-incubated with Eap, the addition of Fn resulted in an even stronger increase of *S. carnosus* internalization compared to that of Eap alone.

Based on our findings presented here, we propose the following model for the Eap-mediated internalization of *S. aureus* and other bacteria into ECs, schematically depicted in Fig. 5.

Secreted Eap stimulates PDI on ECs, which leads to an increased activation of integrins such as $\alpha_5\beta_1$-integrin, thereby augmenting the binding of Fn, which is subsequently bound by *S. aureus* cells via FnBPs and/or cell wall localized Eap. In the bloodstream, it is likely that *S. aureus* FnBPs are already decorated by Fn, thereby enhancing the bacterial cell's capacity to bind to the activated integrin via an Fn bridge. This enhanced Fn binding might lead to an increased integrin clustering which results in elevated uptake of *S. aureus* into the host cell. Eap may also exert additional internalization-enhancing effects because this adhesin can bind to *S. aureus* and Fn and might therefore facilitate the binding of *S. aureus* to the $\alpha_5\beta_1$-integrin via an Fn bridge in addition to FnBPs (29). In the case of *S. carnosus* and other bacteria that are usually only weakly taken up by host cells, the Eap-driven enhanced integrin clustering presumably induces more vivid internalization activity of ECs, accompanied by Eap bridges between the bacterium and Fn-bound $\alpha_5\beta_1$-integrin which enable bacterial uptake (Fig. 5).

While PDI is specifically stimulated by Eap, our finding that the cellular uptake of *S. aureus* was blocked more potently by nonspecific than by specific PDI inhibitors implies important additional contributions by other thiol isomerases in the uptake

process. This concept is actually not surprising considering that integrins, which are of great importance for *S. aureus* uptake, are regulated by specific thiol switches and dithiol-disulfide exchanges (55). Other thiol isomerases expressed in HMEC-1 include ERp5, ERp57, and EndoPDI (65). Consequently, in the future it will be essential to investigate the relative roles of other thiol isomerases in the uptake of *S. aureus* into host cells, and in staphylococcal infections in general, both *in vitro* and *in vivo*. Also, the precise mechanism by which Eap interacts with PDI remains to be elucidated.

For several years, the therapeutic benefit of PDI inhibitors has been investigated in the context of thrombosis and hypercoagulability, from *in vitro* experiments to successful multi-center phase II trials (66–68). The use of PDI inhibitors is also being discussed as a candidate approach for the treatment of cancer (69), and dengue virus infection (70). Our results add another intriguing possibility of therapeutically targeting PDI to modulate the initiation and/or course of *S. aureus* infectious diseases.

## MATERIALS AND METHODS

**Materials.** HMEC-1 (CRL-3243) and EA.hy926 cells (CRL-2922) were purchased from ATCC (LGC Standards GmbH, Germany). ATN161, EDTA-trypsin, HEPES buffer, fetal bovine serum (FBS), L-glutamine, oxidized glutathione (GSSG), eosin isothiocyanate, yeast RNA, scrambled RNase A, hydrocortisone, bacitracin, and 5,5′-dithiobis-(2-nitrobenzoic acid) were from obtained Biochrom/Merck. Human serum albumin (HSA) was obtained from Kedrion Biopharma. MCDB131 medium, Alexa Fluor 488 C5 Maleimide, Lipofectamine RNAiMax, fluorescein-5-isothiocyanate (FITC-celite/isomer), and a bicinchoninic acid protein assay kit were acquired from Thermo Fisher Scientific. Dulbecco's modified Eagle's medium (DMEM), Opti-MEM, hypoxanthine-aminopterin-thymidine (HAT), and epidermal growth factor (EGF) were purchased from Gibco/Thermo Fisher. Accutase was obtained from PromoCell. Rutin trihydrate (quercetin-3-rutinoside) was obtained from Carl Roth. Inhibitory monoclonal antibody anti-PDI RL90 (ab2792) was obtained from Abcam, and RL77 (MA3-018) from Invitrogen/Thermo Fisher. Antibody against activated anti-integrin $\beta_1$ (clone HUTS-4, FCMAB389F) was obtained from Merck. Human P4HB siRNA (SMARTPool ON-TARGETplus human P4HB, L-003690-00-0005) and control siRNA (ON-TARGETplus nontargeting siRNA, D-001810-10-05) was purchased from Dharmacon/Horizon Discovery. Lysostaphin was acquired from AMBI PRODUCTS LLC; fibronectin was obtained from Roche.

**Cell culture.** HMEC-1 (ATCC CRL-3243, immortalized human microvascular endothelial cells) (71) and EA.hy926 (ATCC CRL-2922, a human endothelial hybrid cell line derived by the fusion of human umbilical vein endothelial cells with human cell line A549) (72) were used. The splitting of cells was adapted to their growth speed. For experiments, cells were grown in 12- or 24-well plates (Corning Costar) for 2 to 3 days to confluence in cell-dependent growth medium at 37°C and 5% $CO_2$. EA.hy926 cells were maintained in DMEM supplemented with 10% FBS and 1× HAT medium. HMEC-1 cells were cultivated in MCDB131 base medium completed with 10 ng/mL EGF, 1 $\mu$g/mL hydrocortisone, 10 mM glutamine, and 10% FBS.

**Bacterial strains and culture.** The bacterial strains *S. aureus* 8325-4 (73), Newman (74), and mAH12 (28) and *Staphylococcus carnosus* TM300 (75) were cultivated in tryptic soy broth (TSB) under shaking conditions at 37°C for 15 to 17 h. Bacteria were washed once with phosphate-buffered saline (PBS), adjusted to an optical density at 578 nm ($OD_{578}$) of 1 in TSB + 20% glycerol, and stored at −20°C until use. To determine the number of CFU, bacterial suspensions were plated in serial dilutions on blood agar plates and incubated overnight at 37°C.

**Eap.** For this study, we used two different Eap proteins: recombinant Eap and bacterium-extracted (native) Eap. Recombinant Eap from *S. aureus* Newman without signal peptide was expressed in *E. coli* M15 vector pQE30 and purified as previously described (51). Bacterium-extracted Eap was obtained from *S. aureus* as previously described (76), with the modification that *S. aureus* Newman Δ*sigB* (77) was used. Protein concentrations and product purity were checked by SDS-PAGE and a Bradford assay, respectively. Various assays (invasion of *S. carnosus*, determination of free ecto-sulfhydryls, binding of Fn-FITC to cells) were performed with Eap from both sources to show that Eap protein from different origins did not produce different results. These additional experiments are presented in the supplemental material. The Eap used in each experiment is indicated in the figure captions.

**Measurement of free ecto-sulfhydryls.** The cells were grown to confluence in 12-well cell culture plates at 37°C and 5% $CO_2$. After a washing step with HEPES-Tyrode buffer, host cells were incubated with Eap (1.25 to 20 $\mu$g/mL) for 30 min at 37°C in invasion medium (host cell-dependent basal medium containing 1% HSA and 10 mmol/L HEPES). For inhibitory experiments, cells were pre-incubated with the anti-PDI antibody RL90 (10 $\mu$g/mL) for 15 min before stimulation with Eap. Cells were washed with HEPES-Tyrode buffer and incubated with the thiol-reactive dye Alexa Fluor 488 $C_5$ Maleimide (600 nM) in HEPES-Tyrode buffer for 30 min at 37°C. After washing, cells were detached with accutase, diluted in HEPES-Tyrode buffer, and analyzed by flow cytometry using a BD Accuri C6 and related software.

Instead of Eap, HMEC cells were also incubated with 5% or 10% culture supernatants of different bacterial strains. For this purpose, *S. aureus* 8325-4, Newman, or mAH12 or *S. carnosus* TM300 were cultured in TSB under shaking conditions at 37°C for 15 to 17 h. Bacteria were removed by centrifugation and the culture supernatant was sterile-filtered. The supernatant was added to the invasion medium, and the cells were incubated in this medium at 37°C for 30 min.

**PDI activity assay.** PDI reductase activity was measured as previously described (78), with minor modifications. The PDI substrate di-E-GSSG was prepared by incubating eosin isothiocyanate with GSSG in 0.5 sodium phosphate buffer (2 mM EDTA [pH 8.8]) overnight at room temperature in the dark and separation of the mixture with a Sephadex G-25 column. Fractions with a $20\times$ increase in fluorescence (excitation wavelength [$\lambda_{ex}$] = 485 nm; emission wavelength [$\lambda_{em}$] = 538 nm) were pooled and stored at $-80°C$ in the dark until use.

Soluble PDI (200 nM), trypsinized endothelial cells, or 50 $\mu$g/mL Fn was incubated with Eap (2.5 to 20 $\mu$g/mL, or just 20 $\mu$g/mL) for 30 min, di-E-GSSG was added, and fluorescence was measured in a Fluoroskan-Ascent microplate fluorometer (Thermo Fisher Scientific; $\lambda_{ex}$ = 485 nm; $\lambda_{em}$ = 538 nm). For inhibitory experiments, detached ECs were co-incubated with Eap and DTNB (10 mM), bacitracin (10 mM), or anti-PDI antibody RL90 (10 $\mu$g/mL).

Renaturation of scrambled RNase A as a measure of thiol isomerase activity was assayed according to the methods of Hillson et al. (79). Before the addition of scrambled RNase A, Eap was pre-incubated with soluble PDI for 30 min. The PDI was incubated with scrambled RNase A for 20 min.

**Internalization and adherence assay.** Cells were grown to confluence in 12-well cell culture plates at 37°C and 5% $CO_2$ and washed with PBS before adding 400 $\mu$L of invasion medium. Afterwards, cells were stimulated with Eap (20 $\mu$g/mL) for 30 min at 37°C in preparation for infection.

For inhibitory experiments, cells were pre-incubated with ATN161 (10 $\mu$M), DTNB (10 mM), bacitracin (10 mM), rutin (60 nM), and anti-PDI antibody RL90 (10 $\mu$g/mL) or RL77 (1:100) for 30 min before stimulation with Eap. Since bacitracin also acts as an antibiotic targeting cell wall synthesis in Gram-positive bacteria, host cells were washed before the addition of bacteria in the bacitracin experiments. DNTB and rutin were diluted in DMSO.

Host cells were infected at a multiplicity of infection (MOI) of 50. After a washing step, extracellular bacteria were eliminated by lysostaphin (20 mg/mL) in basal medium for 30 min. For the adherence assay, this step was omitted. Thus, these numbers indicate the bacteria which still adhered after the 1-h incubation plus the bacteria which had been internalized by the host cells but had attached to the cells prior to uptake.

To determine the number of CFU, cells were washed with PBS and incubated with ice-cold water at 4°C for 10 min to lyse host cells and recover the living intracellular bacteria. Cell lysates were mechanically detached and plated in serial dilutions on a blood agar plate and incubated overnight at 37°C. Subsequently, the number of intracellular bacteria was determined by plate counting.

**siRNA assay.** Cells were seeded in 12-well plates in culture medium the day before transfection. After reaching approximately 60% confluence, cells were transfected with P4HB siRNA or negative-control siRNA (10 or 20 pM [see Figure legends] final siRNA were used per well) using Lipofectamine RNAiMAX reagent according to the manufacturer's instructions; RNAiMAX reagent and siRNA were diluted in Opti-MEM before being added to the cells. Cells were incubated at 37°C and 5% $CO_2$ for 2 days before the experiment. Knockdown efficiency was confirmed by Western blotting.

**Western blotting.** After treatment, total cellular protein from HMEC-1 or EA.hy926 cells was extracted using RIPA lysis buffer (10 mM Tris-HCl, 1 mM EDTA, 1% Triton X-100, 0.1% sodium deoxycholate, 0.1% SDS, 140 mM NaCl) containing 1 mM phenylmethanesulfonyl fluoride. Cell lysates were cleared of debris by centrifugation, and the protein concentrations of supernatants were determined using a bicinchoninic acid protein assay kit. Protein in cell lysates was separated by SDS-PAGE (10% pre-cast polyacrylamide gel; Bio-Rad) under reduced conditions and analyzed by Western blotting (anti-P4HB, mouse monoclonal; ab2792; Abcam), anti-GAPDH (glyceraldehyde-3-phosphate dehydrogenase) antibody (mouse monoclonal; G8795; Merck), and a Pageruler Plus Prestained Protein Ladder, 10 to 250 kDa (26619; Thermo Fisher Scientific). Images of Western blots were obtained with a ChemiDoc touch imaging system, and quantification was performed with Image Lab 6.1 software (both Bio-Rad).

**$\beta_1$-Integrin activation.** Cells grown to confluence were washed once with HEPES-Tyrode buffer and subsequently detached with accutase. After a centrifugation step, cells from one well of a 24-well plate were taken up in 100 $\mu$L of HEPES-Tyrode buffer + 1mM $MgCl_2$ to stabilize the confirmation change of integrin (80) and incubated with Eap (20 and 50 $\mu$g/mL) for 30 min at room temperature. When bacitracin or PDI blocking antibody RL77 (1:100) was used, it was added to the cells 15 min before the addition of Eap. After incubation with Eap, a washing step was omitted and the FITC-labeled antibody against activated anti-integrin $\beta_1$ (clone HUTS-4, FCMAB389F, Merck; 4 $\mu$L of the 1:5 diluted antibody solution per 100 $\mu$L) was added directly. After another 30-min incubation at 4°C, the samples were fixed with 4% formaldehyde for 15 min at room temperature. After washing with HEPES-Tyrode buffer, the percentage of antibody-labeled cells was determined by flow cytometry (BD Accuri C6).

**Binding of FITC-coupled Fn to HMEC-1.** The conjugation of Fn with FITC via FITC-celite was performed according to the methods of Xia et al. (81). HMEC-1 or EA.hy926 cells, grown to confluence, were washed with HEPES-Tyrode buffer and stimulated with Eap (20 and 50 $\mu$g/mL) for 30 min at 37°C. Cells were incubated with bacitracin (10 mM) 30 min before the addition of Eap. After another washing step with HEPES-Tyrode-buffer, cells were incubated with 50 $\mu$g/mL Fn-FITC for 30 min, washed again with HEPES-Tyrode buffer, and detached with accutase. Finally, cells were diluted in buffer and analyzed by flow cytometry (BD Accuri C6).

**Statistical analysis.** The results are expressed as mean $\pm$ standard deviation of at least 3 independent experiments. Statistical analyses were performed with Prism (GraphPad Software) using a two-tailed unpaired $t$ test or one-way analysis of variance (ANOVA), followed by Bonferroni or Dunnett's multiple-comparison procedure versus the control as appropriate, or by using two-way ANOVA followed by Bonferroni post-test. $P < 0.05$ was considered statistically significant.

**Data availability.** The data that support the findings of this study are openly available at figshare at https://doi.org/10.6084/m9.figshare.21191317.

## SUPPLEMENTAL MATERIAL

Supplemental material is available online only.

**SUPPLEMENTAL FILE 1**, PDF file, 0.6 MB.

## ACKNOWLEDGMENTS

We thank M. Brück, B. Schuhen, and A. Schulte for excellent technical assistance.

This work was funded by the Deutsche Forschungsgemeinschaft (DFG; German Research Foundation), project ID 194468054–SFB 1009, projects A09 and B01. The funders had no role in study design, data collection and analysis, decision to publish, or preparation of the manuscript.

B.E.K., S.N., J.A.E., A.B., and M.He. conceived and designed the experiments. M.L., A.B., S.N., K.J., and C.R. performed the experiments. M.L., A.B., S.N., and B.E.K. analyzed the data. M.Hu. and M.B. provided Eap. M.Hu. provided *S. aureus* Eap knockout mutant mAH12. S.N., M.L., A.B., A.C.F., and B.E.K. wrote the paper. All authors have read and corrected the paper and agreed to the final version of the manuscript.

We declare no competing interests.

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
