## [Reviewer comments · Microbiology Spectrum]

Microbiology Spectrum

Protein disulfide isomerase and extracellular adherence protein (Eap) cooperatively potentiate staphylococcal invasion into endothelial cells

Marleen Leidecker, Anne Bertling, Muzaffar Hussain, Markus Bischoff, Johannes Eble, Anke Fender, Kerstin Jurk, Christine Rumpf, Mathias Herrmann, Beate Kehrel, and Silke Niemann

Corresponding Author(s): Silke Niemann, University Hospital Münster

Review Timeline:

Submission Date:	September 26, 2022
Editorial Decision:	October 25, 2022
Revision Received:	January 30, 2023
Accepted:	March 9, 2023

Editor: Ana-Maria Dragoi

Reviewer(s): Disclosure of reviewer identity is with reference to reviewer comments included in decision letter(s). The following individuals involved in review of your submission have agreed to reveal their identity: Jérôme Josse (Reviewer #1); Stanimir Ivanov (Reviewer #2)

Transaction Report:

DOI: <https://doi.org/10.1128/spectrum.03886-22>

October 25, 2022

Dr. Silke Niemann
University Hospital Münster
Institute of Medical Microbiology
Münster
Germany

Re: Spectrum03886-22 (Protein disulfide isomerase and extracellular adherence protein (Eap) cooperatively potentiate staphylococcal invasion into endothelial cells)

Dear Dr. Silke Niemann:

The reviewers finished their initial evaluations and while both agree that the study is mostly well done, they have concerns regarding some of the data interpretation. The reviewers also point out that a couple of the experimental approaches and the use of bacteria strains should be better explained. We therefore invite you to submit a revised version with the modifications suggested by the reviewers.

Link Not Available

Sincerely,

Ana-Maria Dragoi

Journals Department
Reviewer comments:

Reviewer #1 (Comments for the Author):

Leidecker et al did a good job studying how exogenous eap can stimulate PDI to enhance the uptake of *S. aureus* and *S. carnosus* inside endothelial cells. They used various methods to demonstrate this mechanism. However, I have some questions regarding the interpretation of the results.

As *S. carnosus* does not produce eap, it seems logical to use exogenous eap to see if eap of *S. aureus* could stimulate the uptake of non-eap producer staphylococci in a context of co-infection. However, it is not clear for me if eap is washed out before adding the bacteria.

If eap is washed out before adding bacteria and as *S. carnosus* does not seem to invade host cells through a FnBP/fibronectin/integrin $\alpha 5\beta 1$ mechanism, eap should stimulate uptake through an alternative invasion pathway as eap is not present and cannot link *S. carnosus* to fibronectin (to invade through the $\alpha 5\beta 1$ integrin pathway).

It could be interesting to verify that the increased uptake of *S. carnosus* in the presence of eap is $\alpha 5\beta 1$ integrin dependent. Regarding *S. aureus* experiments, I am questioning if endogenous eap (produced by *S. aureus* during the infection) could also enhance the uptake of *S. aureus* by the host cells. Indeed, results from HMEC-1 invasion (Fig S3A) showed that bacitracin has a reducing effect on *S. aureus* uptake without stimulation with eap. With EAhy926 cells, difference of uptake with or without bacitracin in absence of eap stimulation was not significant but a trend is observed.

In Figure 2A, the order of conditions is not optimal. Bacitracin alone should be next control and then you have eap alone and eap + bacitracin. So you can compare effect of bacitracin without exogenous eap (endogenous eap) and then you can compare in the presence of exogenous.

Moreover, still in Figure 2, I don't understand why it is relevant to perform statistical analysis between internalization with eap and internalization with bacitracin alone. They are two factors that are modulated so you cannot interpretate the results.

It could have been interesting to test a strain with normal or high production of eap and its isogenic mutant deleted for eap to investigate if endogenous production of eap could also stimulate *S. aureus* uptake.

Moreover, are the concentrations of exogenous eap comparable to endogenous production by *S. aureus*? I am curious then, as 20 $\mu\text{g}/\text{mL}$ of exogenous eap increased staphylococci uptake, increased PDI activity but did not increase $\beta 1$ integrin activation, it seems difficult to relate eap stimulation with activation of $\beta 1$ integrin-dependent uptake through PDI activation.

For me, authors need to perform additional experiments (eap mutant and blockage of $\beta 1$ integrin) to validate their interpretation of the results or they need to rework the manuscript to clarify their findings by specifying that staphylococcal uptake is stimulated by exogenous eap and that the uptake could be related to $\beta 1$ integrin pathway but that it was not fully validated.

Reviewer #2 (Comments for the Author):

In this work, Leidecker et al investigate the mechanism by which the extracellular adherence protein (Eap) secreted by *S. aureus* enhances invasion of endothelial cells. The authors provide experimental evidence using a combination of pharmacological and genetic approaches that suggests Eap activates plasma membrane localized PDI and together with bacterial fibronectin-binding proteins promotes bacterial internalization by non-phagocytic cells. While overall the data as presented by the authors supports the model, skepticism is raised by certain data inconsistencies (with their model) and choice of experimental approaches for key experiments. If the following the main deficiencies or unaddressed questions are resolved, I feel the manuscript will be significantly strengthened in both rigor and confidence in the proposed model:

(1) It is unclear whether the *S. aureus* (SA) and *S. carnosus* (SC) strains used in this work produce and secrete Eap. In the various experiments performed, the authors use purified Eap. This is an important point because if the bacteria secrete Eap the question if the Eap is required for invasion in their model is not tested at all. If the strains used in the study secrete Eap, then the authors in the key experiments should either compare WT vs ΔEap strain or use ΔEap mutants +/- purified Eap. Either way, the rationale for choosing particular strains should be clearly stated and justified.

(2) Despite the authors' assertions, Eap-dependent enhancement of SC invasion appears to be largely independent of PDI as is seen in Figures (3F/G/I) where PDI-blocking antibodies (3F-G) and siRNA of PDI (3I) barely reduce invasion when Eap is supplemented. Yes, there is a statistically significant reduction in invasion upon PDI blockade, but that reduction is minor, which demonstrates that Eap facilitates invasion in a manner largely independent of PDI.

(3) It is puzzling to this reviewer why the authors use bacitracin for key experiments (Fig 4) in the paper. Bacitracin use is problematic because (i) it is an antibiotic and (ii) it does not appear to be a good (let alone specific inhibitor for PDI) (PMID: 20477872). Moreover, in the paper bacitracin treatment does NOT phenocopy the data from PDI knockdown. As seen in Fig 4B bacitracin treatment always elicits a much stronger defect as compared to PDI KD, also compare data from Fig 3B to Fig 3F/G/I. Therefore, the experiments in Fig 4A, 4C and 4D should be repeated using PDI knockdown approach. I don't understand why the authors chose to use a questionable inhibitor over a sound genetic approach. They clearly demonstrate that they can reduce PDI expression using siRNA knock down (see Fig3J).

(4) The authors have an assay to measure disulfate activity on the cell surface (Fig 2A). Despite having this assay at their disposal, they never bother to measure if and by how much the various approaches targeting PDI (antibody blockade and siRNA KD) reduce plasma membrane-associated disulfate activity. Experimental data along those lines will instill greater confidence that those interference approaches actually have the predicted effect on PDI enzymatic activity.

Staff Comments:

Preparing Revision Guidelines

Please return the manuscript within 60 days; if you cannot complete the modification within this time period, please contact me. If you do not wish to modify the manuscript and prefer to submit it to another journal, please notify me of your decision immediately so that the manuscript may be formally withdrawn from consideration by Microbiology Spectrum.

Response to Reviewers

Reviewer #1 (Comments for the Author):

Leidecker et al did a good job studying how exogenous eap can stimulate PDI to enhance the uptake of *S. aureus* and *S. carnosus* inside endothelial cells. They used various methods to demonstrate this mechanism.

Response: Thank you very much for this supportive comment.

However, I have some questions regarding the interpretation of the results.

1. As *S. carnosus* does not produce eap, it seems logical to use exogenous eap to see if eap of *S. aureus* could stimulate the uptake of non-eap producer staphylococci in a context of co-infection. However, it is not clear for me if eap is washed out before adding the bacteria.

If eap is washed out before adding bacteria and as *S. carnosus* does not seem to invade host cells through a FnBP/fibronectin/integrin $\alpha 5\beta 1$ mechanism, eap should stimulate uptake through an alternative invasion pathway as eap is not present and cannot link *S. carnosus* to fibronectin (to invade through the $\alpha 5\beta 1$ integrin pathway).

Response: In most experiments, exogenous Eap was not washed away. This was however done whenever bacterial internalisation was investigated in the presence of bacitracin. Here, the cells were pre-incubated with bacitracin for 30 min in all approaches, i.e. also in the control cells without bacitracin, followed by the incubation with Eap. Afterwards, a washing step was conducted and the cells were incubated with the bacteria. This is indicated e.g. in lines 176-177, lines 309-312 and lines 484-486.

As our results show (e.g. Fig. 3), bacterial uptake is not noticeably different in the presence or absence of Eap (with or without washing step), indicating no direct modulation on the internalization process per se. Eap can bind to $\alpha 5\beta 1$ integrin during the incubation period, and in the next step *S. aureus* or *S. carnosus* can attach to the Eap. Previous work has also shown that unbound Eap is not required for Eap-driven bacterial invasion of non-professional phagocytic cells.(1) We also refer to this in lines 309-312.

2. It could be interesting to verify that the increased uptake of *S. carnosus* in the presence of eap is $\alpha 5\beta 1$ integrin dependent.

Response: We thank the reviewer for this valuable idea. To test whether the Eap-enhanced uptake of *S. carnosus* TM300 is $\alpha 5\beta 1$ integrin-dependent, we used the integrin inhibitor ATN161 (2). Eap-mediated uptake into HMEC-1 was significantly reduced by ATN161 (new Suppl. Fig. S4B), so we can assume that at least part of the Eap mediated *S. carnosus* uptake is integrin dependent (lines 209-211 and lines 281-283).

3. Regarding *S. aureus* experiments, I am questioning if endogenous eap (produced by *S. aureus* during the infection) could also enhance the uptake of *S. aureus* by the host cells. Indeed, results from HMEC-1 invasion (Fig S3A) showed that bacitracin has a reducing effect on *S. aureus* uptake

without stimulation with eap. With EAhy926 cells, difference of uptake with or without bacitracin in absence of eap stimulation was not significant but a trend is observed.

Response: The reviewer is absolutely right in that both bacitracin and DTNB, i.e. the non-selective thiol isomerase inhibitors reduce the uptake of *S. aureus*. This could be due to the low amount of endogenously produced Eap by *S. aureus* 8325-4 (see also response to point 5), however, even the already very low uptake of *S. carnosus* TM300 is further reduced by bacitracin and especially by DTNB, although *S. carnosus* does not produce Eap at all. We discuss for this reason in lines 372-381 to 373 that we think that further thiol isomerases such as Erp57 or ERp5, which are also expressed by endothelial cells (compare e.g. Smith et al., 2015 (3)), additionally play an important role in general for the uptake process besides PDI. The relative importance of the other thiol isomerases will be the subject of future studies.

4. In Figure 2A, the order of conditions is not optimal. Bacitracin alone should be next control and then you have eap alone and eap + bacitracin. So you can compare effect of bacitracin without exogenous eap (endogenous eap) and then you can compare in the presence of exogenous. Moreover, still in Figure 2, I don't understand why it is relevant to perform statistical analysis between internalization with eap and internalization with bacitracin alone. They are two factors that are modulated so you cannot interpretate the results.

Response: We thank the reviewer for this suggestion and have changed the order of conditions. We have also redone the statistical analysis.

5. It could have been interesting to test a strain with normal or high production of eap and its isogenic mutant deleted for eap to investigate if endogenous production of eap could also stimulate *S. aureus* uptake.

Response: We have added an experiment on the internalisation of *S. aureus* Newman and the mutant strain *S. aureus* mAH12 with a deficient *eap* gene (Hussain et al. 2002 (4)) (new Fig. 1E). The strain mAH12 internalised to a lesser extent than the wild-type strain *S. aureus* Newman. The addition of exogenous Eap restores the uptake of the eap-knock out mutant. The strain Newman is known to have a significantly higher Eap production than the strain *S. aureus* 8325-4 used so far for our experiments. Our results confirm the findings of Haggart et al. 2003 (5) that endogenous Eap production increases uptake or uptake is reduced when it is absent. We were not able to use supernatant from *S. aureus* Newman instead of exogenous Eap for the internalization experiments with *S. carnosus* TM300, because even 5% supernatant led to markedly increased cell death in the host cells at a 1h incubation time. Our additional results are described and discussed in lines 143-147 and lines 269-280 in the revised manuscript.

We also investigated the effect of bacterial supernatants from the bacteria used in our experiments on the abundance of free sulfhydryls. As can be seen in new Fig. 2D, incubation of HMEC-1 cells with bacterial supernatants of *S. carnosus* TM300 and also of *S. aureus* 8325-4 for 30 minutes does not lead to an increase in Alexa Fluor 488 C5 maleimide positive cells, in contrast to incubation with supernatant from *S. aureus* Newman. Our additional results are described and discussed in lines 157-160 and lines 294-301.

6. Moreover, are the concentrations of exogenous eap comparable to endogenous production by *S. aureus*? I am curious

Response: In the publication by Palma et al. 1999 (6), the amount of Eap isolated from 4- and 24-h cultures from the supernatant of *S. aureus* Newman is given as 1.05 (4-h culture) and 6.47 mg/l (24-h culture). Thus, we use a slightly higher concentration for our experiments than that produced by *S. aureus* Newman, but use a concentration of the same order of magnitude. We have now also given this information in the manuscript (lines 260-263).

7. Then, as 20µg/mL of exogenous eap increased staphylococci uptake, increased PDI activity but did not- increase beta 1 integrin activation, it seems difficult to relate eap stimulation with activation of beta1 integrin-dependent uptake through PDI activation.

Response: As we noted in lines 333-339, we attribute the issue of having to use 50 µg/ml Eap for the experiments to a methodological problem: “We could show here that the addition of Eap to HMEC-1 cells also caused an increased activation of β1-integrin, although this effect was more evident at high concentrations (i.e. 50 µg/ml) of this bacterial adhesion protein. However, cell culture conditions at atmospheric O₂ concentration represent a more oxidative environment than occurs in tissue in situ. Also, the sensitivity of the antibody or of the method might be responsible for the required higher Eap concentration.”

For me, authors need to perform additional experiments (**eap mutant and blockage of b1 integrin**) to validate their interpretation of the results or they need to rework the manuscript to clarify their findings by specifying that staphylococcal uptake is stimulated by exogenous eap and that the uptake could be related to beta1 integrin pathway but that it was not fully validated.

Response: We hope that we could convince the reviewer with our new results.

Reviewer #2 (Comments for the Author):

In this work, Leidecker et al investigate the mechanism by which the extracellular adherence protein (Eap) secreted by *S. aureus* enhances invasion of endothelial cells. The authors provide experimental evidence using a combination of pharmacological and genetic approaches that suggests Eap activates plasma membrane localized PDI and together with bacterial fibronectin-binding proteins promotes bacterial internalization by non-phagocytic cells. While overall the data as presented by the authors supports the model, skepticism is raised by certain data inconsistencies (with their model) and choice of experimental approaches for key experiments. If the following the main deficiencies or unaddressed questions are resolved, I feel the manuscript will be significantly strengthened in both rigor and confidence in the proposed model:

Response: We thank the reviewer for his constructive comments and hope that with our new data we can dispel the concerns.

(1) It is unclear whether the *S. aureus* (SA) and *S. carnosus* (SC) strains used in this work produce and

secrete Eap. In the various experiment performed, the authors use purified Eap. This is an important point because if the bacteria secrete Eap the question if the Eap is required for invasion in their model is not tested at all.

If the strains used in the study secrete Eap, then the authors in the key experiments should either compare WT vs Δ Eap strain or use Δ Eap mutants +/- purified Eap. Either way, the rationale for choosing particular strains should be clearly stated and justified.

Response: We have now placed more emphasis on the information regarding Eap expression of the staphylococci used (see e.g. lines 130-135, line 294-297). In addition, we have added an experiment on the internalisation of *S. aureus* Newman and the mutant strain *S. aureus* mAH12 with a deficient eap gene (Hussain et al. 2002 (4)) (new Fig. 1E). The strain mAH12 is less internalised than the wild-type strain *S. aureus* Newman. The addition of Eap significantly increases the uptake of the eap-knock out mutant. The strain Newman is known to have a significantly higher Eap production than the strain *S. aureus* 8325-4 used so far for our experiments. Our results confirm the findings of Hagggar et al. 2003 (5) that endogenous Eap production increases uptake or uptake is reduced when it is absent. Our additional results are described and discussed in lines 143-147 and lines 268-280 in the revised manuscript.

We also investigated the effect of bacterial supernatants from the bacteria used in our experiments on the abundance of free sulfhydryls. As can be seen in new Fig. 2D, incubation of HMEC-1 cells with bacterial supernatants of *S. carnosus* TM300 and also of *S. aureus* 8325-4 for 30 minutes does not lead to an increase in Alexa Fluor 488 C5 maleimide positive cells, in contrast to incubation with supernatant from *S. aureus* Newman. Our additional results are described and discussed in lines 157-160 and lines 294-301.

(2) Despite the authors assertions, Eap-dependent enhancement of SC invasion appears to be largely independent of PDI as is seen in Figures (3F/G/I) where PDI-blocking antibodies (3F-G) and siRNA of PDI (3I) barely reduce invasion when Eap is supplemented. Yes, there is a statistically significant reduction in invasion upon PDI blockade, but that reduction is minor, which demonstrates that Eap facilitates invasion in a manner largely independent of PDI.

Response: *S. carnosus* is only taken up to a very low extent without the addition of Eap, for this reason the number of internalized bacteria is significantly higher in terms of percentage compared to the control than for the internalization of *S. aureus* into EA.hy926 cells. In order to be able to show the control samples (without Eap) as well, we decided to show the Y-axis as a broken axis. This may have given the reviewer the impression that the reduction by siRNA knock down and the blocking antibodies is only minor. For example, in Fig. 3G, the mean value for internalization of *S. carnosus* after Eap addition is 12913%. Blocking the PDI by RL77 reduces the uptake to 2921%. By siRNA (Fig. 3I), internalization is reduced from 23180% (with Eap) to 10889% (Eap and siRNA). Thus, the reduction is not only statistically significant but also higher than the reviewer may have thought. In addition, we also state that our results obtained with the non-selective inhibitors bacitracin and DTNB, which led to an even greater reduction in uptake, suggest to us that other thiol isomerases are also important for internalization (lines 372-377). In addition, in *S. carnosus*, the binding of the bacteria to α 5 β 1 integrin via an Eap-Fn bridge is also important and allows the uptake. We discuss this in lines 364-370.

(3) It is puzzling to this reviewer why the authors use bacitracin for key experiments (Fig 4) in the paper. Bacitracin use is problematic because (i) it is an antibiotic and (ii) it does not appear to be a good (let alone specific inhibitor for PDI) (PMID: 20477872). Moreover, in the paper bacitracin treatment does NOT phenocopy the data from PDI knockdown. As seen in Fig 4B bacitracin treatment always elicits a much stronger defect as compared to PDI KD, also compare data from Fig 3B to Fig 3F/G/I. Therefore, the experiments in Fig 4A, 4C and 4D should be repeated using PDI knockdown approach.

I don't understand why the authors chose to use a questionable inhibitor over a sound genetic approach. They clearly demonstrate that they can reduce PDI expression using siRNA knock down (see Fig3J).

Response: We thank the reviewer for his concerns and hope to address them appropriately in our revised manuscript and in this reply.

We circumvented the problem of bacitracin as an antibiotic by washing away this inhibitor, as well as Eap, before adding the bacteria when using bacitracin. Please see also lines 176-177, lines 310-312 and lines 484-486.

Bacitracin is conventionally used as a PDI inhibitor in many publications. It has been shown that bacitracin binds to free cysteines in the substrate-binding domain of PDI (Dickerhof et al., 2011 (7)). However, the authors of this study also state that bacitracin can also bind to other proteins containing free cysteines. We have seen that both bacitracin and DTNB had a stronger effect in our experiments than the PDI-specific inhibitors or the PDI siRNA knockdown. We therefore discuss in our manuscript that this may be because there are other thiol isomerases involved in the (Eap-mediated) internalization of the bacteria that are also blocked by bacitracin and DTNB (lines 372-377).

However, we also followed the very valuable suggestion of the reviewer and performed further experiments on integrin activation and bacterial internalization using PDI inhibitory antibodies and a PDI siRNA knockdown approach, respectively (new Fig. 4A and B and new Fig. 4G and H). We also obtained significant inhibition of the promoting Eap-effect using these methods. We hope that these novel insights convince the reviewer that Eap has a stimulatory effect on PDI and this can also lead to enhanced bacterial internalization. Our additional results are described and discussed in lines 205-207, lines 229-232 and lines 339-342 in the revised manuscript.

(4) The authors have an assay to measure disulfate activity on the cell surface (Fig 2A). Despite having this assay at their disposal, they never bother to measure if and by how much the various approached targeting PDI (antibody blockade and siRNA KD) reduce plasma membrane-associated disulfate activity. Experimental data along those lines will instill greater confidence that those interference approaches actually have the predicted effect on PDI enzymatic activity.

Response: We thank the reviewer for this comment. In the revised version of our manuscript, we now show that both antibody blockade and siRNA knockdown significantly reduce the Eap-promoted increase in PDI activity as well as reductase activity (new Fig. 2C and G). Our additional results are described and discussed in lines 153-156, lines 164-167 and lines 301-305.

Literature:

1. Bur, S., Preissner, K. T., Herrmann, M., and Bischoff, M. (2013) The Staphylococcus aureus extracellular adherence protein promotes bacterial internalization by keratinocytes independent of fibronectin-binding proteins. *J Invest Dermatol* **133**, 2004-2012
2. Cheng, W. H., Lee, K. Y., Yu, M. C., Chen, J. Y., Lin, C. H., and Chen, B. C. (2021) Pref-1 induced lung fibroblast differentiation by hypoxia through integrin alpha5beta1/ERK/AP-1 cascade. *Eur J Pharmacol* **909**, 174385
3. Smith, C. L., Shah, C. M., Kamaludin, N., and Gordge, M. P. (2015) Inhibition of thiol isomerase activity diminishes endothelial activation of plasminogen, but not of protein C. *Thromb Res* **135**, 748-753
4. Hussain, M., Hagggar, A., Heilmann, C., Peters, G., Flock, J. I., and Herrmann, M. (2002) Insertional inactivation of Eap in Staphylococcus aureus strain Newman confers reduced staphylococcal binding to fibroblasts. *Infect Immun* **70**, 2933-2940
5. Hagggar, A., Hussain, M., Lönnies, H., Herrmann, M., Norrby-Teglund, A., and Flock, J. I. (2003) Extracellular adherence protein from Staphylococcus aureus enhances internalization into eukaryotic cells. *Infect Immun* **71**, 2310-2317
6. Palma, M., Hagggar, A., and Flock, J. I. (1999) Adherence of Staphylococcus aureus is enhanced by an endogenous secreted protein with broad binding activity. *J Bacteriol* **181**, 2840-2845
7. Dickerhof, N., Kleffmann, T., Jack, R., and McCormick, S. (2011) Bacitracin inhibits the reductive activity of protein disulfide isomerase by disulfide bond formation with free cysteines in the substrate-binding domain. *FEBS J* **278**, 2034-2043

March 9, 2023

Dr. Silke Niemann
University Hospital Münster
Institute of Medical Microbiology
Münster
Germany

Re: Spectrum03886-22R1 (Protein disulfide isomerase and extracellular adherence protein (Eap) cooperatively potentiate staphylococcal invasion into endothelial cells)

Dear Dr. Silke Niemann:

Your manuscript has been accepted, and I am forwarding it to the ASM Journals Department for publication. You will be notified when your proofs are ready to be viewed.

Sincerely,

Ana-Maria Dragoi
Editor, Microbiology Spectrum
